# Neutrophil to Lymphocyte Ratio in Oropharyngeal Squamous Cell Carcinoma: A Systematic Review and Meta-Analysis [note 1]

**DOI:** 10.3390/cancers15030802

**Published:** 2023-01-28

**Authors:** Juan P. Rodrigo, Mario Sánchez-Canteli, Asterios Triantafyllou, Remco de Bree, Antti A. Mäkitie, Alessandro Franchi, Henrik Hellquist, Nabil F. Saba, Göran Stenman, Robert P. Takes, Cristina Valero, Nina Zidar, Alfio Ferlito

**Affiliations:** 1Department of Otolaryngology, Hospital Universitario Central de Asturias and Instituto de Investigación Sanitaria del Principado de Asturias (ISPA), Instituto Universitario de Oncología del Principado de Asturias, University of Oviedo, 33011 Oviedo, Spain; 2Centro de Investigación Biomédica en Red de Cáncer (CIBERONC), Instituto de Salud Carlos III, 28029 Madrid, Spain; 3Department of Pathology, Liverpool Clinical Laboratories, School of Dentistry, University of Liverpool, Liverpool L3 5PS, UK; 4Department of Head and Neck Surgical Oncology, University Medical Center Utrecht, University of Utrecht, 3584 CX Utrecht, The Netherlands; 5Department of Otorhinolaryngology—Head and Neck Surgery, University of Helsinki and Helsinki University Hospital, 00290 Helsinki, Finland; 6Department Translational Research and of New Technologies in Medicine and Surgery, University of Pisa, 56126 Pisa, Italy; 7Faculty of Medicine and Biomedical Sciences (FMCB), University of Algarve, Algarve Biomedical Center Research Institute (ABC-RI), 8005-139 Faro, Portugal; 8Department of Hematology and Medical Oncology, Emory University School of Medicine, Atlanta, GA 30322, USA; 9Sahlgrenska Center for Cancer Research, Department of Laboratory Medicine, University of Gothenburg, SE-405 30 Gothenburg, Sweden; 10Department of Otolaryngology-Head and Neck Surgery, Radboud University Medical Center, 6500 AB Nijmegen, The Netherlands; 11Otorhinolaryngology Department, Hospital de La Santa Creu I Sant Pau, Universitat Autònoma de Barcelona, 08041 Barcelona, Spain; 12Institute of Pathology, Faculty of Medicine, University of Ljubljana, 1000 Ljubljana, Slovenia; 13Coordinator of the International Head and Neck Scientific Group, 35125 Padua, Italy

**Keywords:** oropharyngeal squamous cell carcinoma, meta-analysis, neutrophil-to-lymphocyte ratio, prognosis

## Abstract

**Simple Summary:**

Neutrophil-to-lymphocyte ratio (NLR) in peripheral blood samples has been associated with prognosis in several cancers, including head and neck cancer, but in oropharyngeal carcinomas its prognostic value, especially in relation to human papillomavirus (HPV) infection, has been little studied. This meta-analysis, including studies on the prognostic value of NLR in oropharyngeal carcinoma, shows that an elevated pretreatment NLR is associated with a worse prognosis in oropharyngeal cancer, regardless of the type of treatment performed, but this prognostic value appears to be specific to HPV-positive oropharyngeal carcinomas. NLR could be used as an affordable prognostic marker in this type of cancer.

**Abstract:**

Neutrophil-to-lymphocyte ratio (NLR) has been associated with survival in various cancers, including head and neck cancer. However, there is limited information on its role in oropharyngeal squamous cell carcinomas (OPSCC) according to HPV status. This prompted the present meta-analysis. Studies were selected when the prognostic value of NLR prior to treatment was evaluated in OPSCC patients, the cutoff value of NLR was available, and the prognostic value of NLR was evaluated by time-to-event survival analysis. A total of 14 out of 492 articles, including 7647 patients, were analyzed. The results showed a worse prognosis for the patients with a high NLR: The combined hazard ratios (HR) for overall survival (OS) in patients with an elevated NLR was 1.56 (95% confidence interval (CI) 1.21–2.02; *p* = 0.0006), for disease-free survival was 1.52 (95% CI 1.34–1.73; *p* < 0.00001), and for recurrence-free survival was 1.86 (95% CI 1.50–2.30; *p* < 0.00001). This worse prognosis of high NLR was exclusive of HPV-positive patients: HR for OS in the HPV-positive subgroup was 4.05 (95% CI 1.90–8.62 (*p* = 0.0003), and in the HPV-negative subgroup 0.92 (95% CI 0.47–1.80; *p* = 0.82). The prognosis of NLR was not influenced by treatment: The HR for OS for patients treated with radiotherapy/chemoradiotherapy (RT/CRT) was 1.48 (95% CI 1.09–2.01; *p* = 0.01), and for patients treated with surgery (±RT/CRT) was 1.72 (95% CI 1.08–2.72; *p* = 0.02). In conclusion, an elevated NLR relates to worse outcomes in patients with HPV-positive OPSCC.

## 1. Introduction

Oropharyngeal squamous cell carcinoma (OPSCC) represents a subgroup of head and neck cancers (HNSCCs) and includes tumors arising from the tonsils, base of the tongue, posterior pharyngeal wall, and soft palate [1,2]. More than 70% of OPSCCs in developed countries are driven by a human papilloma virus (HPV) infection [2]. HPV driven OPSCC has distinct demographical, pathological, and molecular features and a more favorable prognosis compared with non-HPV related OPSCC, usually associated with tobacco and alcohol consumption. HPV-driven OPSCC occur in younger patients with no or less history of tobacco and alcohol consumption and show a better response to treatment. These tumors are caused by HPV infection and are defined by the presence of high-risk types of HPV in tumor cells. The viral E6 and E7 oncoproteins inactivate the cell cycle regulatory proteins p53 and pRB, respectively, and are required for malignant transformation [3,4].

The role of host immunity in the natural history of neoplasia has been intensely investigated. Pertinent immune responses are often manifested by infiltration of tumor stroma and parenchyma by lymphomononuclear immune cells attempting to monitor and protect against tumor growth [5,6]. Evasion of the immune response is an important part of cancer, expected to adversely affect tumor prognosis. Various mechanisms of immune evasion have been characterized and explored as key targets for anti-cancer immunotherapy [7]. 

Host inflammatory responses to cancer are associated with poor tumor-specific prognosis [8,9]. In the early stages of tumor development, there is an anti-tumor immune response, but as the tumor progresses it develops immune evasion mechanisms while triggering an inflammatory response that has pro-tumorigenic effects. This altered inflammatory state acts locally, but also triggers systemic inflammation [10]. The proportions of circulating inflammatory cells obtained from routine blood counts, including the white blood cell count, lymphocyte count, neutrophil count, and platelet count, are used as indicators of systemic inflammation. In addition, to better assess systemic inflammation scores or ratios of some of these parameters have been evaluated and have been found to be correlated with the prognosis of numerous pathologies, including cancer [10].

The neutrophil-to-lymphocyte ratio (NLR) in peripheral blood, regarded as the balance between systemic inflammation and immunity, has proven associated with patient survival in cancers of larynx, liver, stomach, esophagus and breast, and soft-tissue sarcomas [9,11,12]. While there is an increasing number of studies reporting poorer prognosis in patients with elevated NLR in HNSCC [13,14,15,16,17,18,19,20,21,22,23,24,25,26,27,28,29,30,31,32,33,34,35,36,37,38,39,40,41,42], these usually include a limited number of patients from different anatomical locations and there is no consensus on the optimal cut off value of NLR. In addition, while some studies have focused on OPSCC, a meta-analysis assessing the prognostic significance of NLR in this tumor type specifically and based on HPV status has not been undertaken. Therefore, the present review critically analyzes the literature to explore the relationship between NLR prior to treatment and outcome in OPSCC, and in relation to the HPV status.

## 2. Materials and Methods

### 2.1. Data Source

The Preferred Reporting Items for Systematic Review and Meta-Analyses (PRISMA) were used to conduct a systematic review of the literature [43]. This systematic review is registered in Open Science (identifier: DOI 10.17605/OSF.IO/6P4DE). The search strategy aimed to include all articles concerning the role of NLR prior to treatment in OPSCC. A PubMed internet search updated to 15 March, 2022, was performed for publications in English language between 1990 and 2022 using the following key words in the title or abstract: “neutrophil lymphocyte ratio” coupled with “Head and Neck Squamous Cell Carcinoma” OR “Head and Neck Cancer” OR “Oropharyngeal Squamous Cell Carcinoma” OR “Oropharynx Cancer” (Appendix A).

The search results were independently reviewed by two of the authors (JPR and MSC) for potentially eligible studies. When follow-up data and outcomes of the role of NLR prior to treatment in OPSCC were mentioned in the abstract, the full text article was perused. Review articles were also perused in full. References from any fully perused articles were cross-checked to ensure appropriate inclusion in the present review (Figure 1). Disagreement over eligibility of inclusion was resolved by consensus.

### 2.2. Study Selection

Studies that satisfy the following inclusion criteria were selected for the study: (1) The prognostic significance of NLR was evaluated in patients with OPSCC; (2) NLR was obtained prior to treatment; (3) the cutoff value of NLR was available; (4) the prognostic value of pretreatment NLR was analyzed by time-to-event survival analysis with overall survival (OS), disease-free survival (DFS), locoregional control (LRC) and/or recurrence free survival (RFS); and (5) original articles published in English between January 1990 and 15 March 2022.

Studies were excluded if (1) information about prognostic accuracy was lacking or with insufficient data for assessing Hazard ratios (HR) with a 95% confidence interval (95% CI) and (2) corresponded with letters to the editor, case reports, non-clinical studies, and conference abstracts.

### 2.3. Data Extraction

Two authors (Juan P. Rodrigo and Mario Sánchez-Canteli) independently extracted the relevant information in each selected paper and disagreements were resolved by consensus. Prespecified data from each article were recorded as follows: Name of the first author, year of publication, country, sample size, design of study, median age, median follow-up period, HPV status, clinical stage, treatment method, cut-off value of NLR, method of obtaining the cut-off value, survival analysis and analysis method. In the case of univariate and multivariate data being available, preference was given to the latter.

### 2.4. Quality Assessment

The quality of eligible studies was independently evaluated by two authors (Juan P. Rodrigo and Mario Sánchez-Canteli) via the Newcastle-Ottawa Scale (NOS), using three parameters with a maximum of 9 points: comparability (0–4 points), selection (0–2 points), and outcome confirmation (0–3 points) [44]. Scores > 6 points and ≤6 points indicated high and low-quality articles, respectively (Appendix A).

### 2.5. Data Synthesis

The meta-analysis was undertaken using Review Manager 5.4. Subgroup analysis stratified by HPV status was conducted. Heterogeneity was assessed using *I*^2^ statistic, with <25% regarded as low level, 25–50% as moderate level, and >50% as high level. HRs were used to describe the risk of event (OS, DFS, RFS, LRC) for high versus low NLR. If no significant heterogeneity (*I*^2^ < 50%) was detected, the fixed model was subsequently used. If there was significant heterogeneity, a random effect model was implemented. Forest plots and funnel plots were employed to test the overall effect and the publication bias, respectively. All the tests were two sided with a significance level of *p* < 0.05.

## 3. Results

### 3.1. Literature Search and Included Studies

Following the search criteria, 492 papers were initially identified but only 193 were related to the topic of the study. After sorting and removing duplicates, the abstracts of the remaining papers were reviewed, 26 of which were retrieved and reviewed in detail. Twelve studies that did not satisfy the inclusion criteria were discarded [27,28,29,30,31,32,33,34,35,36,37,38]. Of these, nine were excluded because they did not report the outcome of OPSCC separately from other locations of HNSCC [27,28,29,30,31,32,34,37,38], and three because they did not report the NLR value [33,35,36]. Fourteen studies comprising 7647 patients with OPSCC were eventually selected in the analysis [13,14,15,16,17,18,19,20,21,22,23,24,25,26] (Figure 1). All studies were retrospective and of a high-quality with scores > 6 points on the NOS. Table 1 summarizes the main features of the selected studies and Appendix A summarizes the key findings.

### 3.2. Method of Obtaining the Cut-Off Value of NLR

All NLR analyses were from peripheral blood sampled prior to any treatment. Each of the articles had a different way of establishing the cut-off values for high and low NLR, the values ranging from 2.0 to 5.0 (mean 3.6 and median 3.4). Two studies defined a high NLR with a value over 5.0 according to a review of literature [23,25]. In six studies, an ROC analysis was used to determine the cut-off value, which varied between 2.009 and 4.7 [14,15,18,19,20,21]. Three studies established the cut-off value according to the median (3.0) value of NLR [16,17,26]. Another study [13] established the value according to the top tertile value of NLR (3.4 for OS and 3.7 for DFS), and another one [24] according to the top quartile value of NLR (4.68). In one study, the value was not available [16].

### 3.3. Impact of NLR on Survival in All Patients

The prognostic value of NLR on survival in all patients, independently of HPV status, was assessed on the 14 eventually selected studies [13,14,15,16,17,18,19,20,21,22,23,24,25,26], although not all included OS, DFS, RFS and LRC data (Figure 2). All showed a survival disadvantage for cases with high NLR. The pooled meta-analysis showed a survival disadvantage for high NLR (pooled HR 1.56 (95% CI 1.21–2.02), *p* = 0.0006 for OS; 1.52 (95% CI 1.34–1.73), *p* < 0.0001, for DFS; 1.86 (95% CI 1.50–2.30), *p* < 0.0001, for RFS; and 1.85 (95% CI 1.30–2.63), *p* = 0.007, for LRC). All studies reported the NLR value prior to treatment, but it is re-emphasized that not all used the same cut-off point.

### 3.4. Impact of NLR on Survival in HPV-Positive Patients

The prognostic value of NLR on survival in HPV-positive patients was assessed on three of the eventually selected studies [14,20,25] comprising 398 HPV-positive patients, although not all included DFS data (Figure 3A,B). All showed a survival disadvantage for cases with high NLR. The pooled meta-analysis showed a survival disadvantage for high NLR (pooled HR 4.05 (95% CI 1.9–8.62), *p* = 0.0003 for OS; and 4.76 (95% CI 1.81–12.50), *p* = 0.002, for DFS).

### 3.5. Impact of NLR on Survival in HPV-Negative Patients

The prognostic value of NLR on survival in HPV-negative patients was assessed on three of the eventually selected studies [15,16,20] comprising 292 HPV-negative patients, although not all included DFS data (Figure 3C,D). The pooled meta-analysis did not show differences for high NLR (pooled HR 0.92 (95% CI 0.47–1.80), *p* = 0.82 for OS; and 1.88 (95% CI 0.99–3.57), *p* = 0.05, for DFS).

### 3.6. Impact of NLR on Survival in All Patients by Treatment

According to the type of cancer treatment, the combined analysis of eight studies [15,19,20,21,24,25,26] showed that patients treated exclusively with radiotherapy (R) or chemoradiotherapy (CR) with high NLR had a survival disadvantage (pooled HR 1.48 (95% CI 1.09–2.01), *p* = 0.01, for OS) (Figure 4A). The combined analysis of the remaining six studies [13,14,16,17,18,23] that included patients with mixed primary treatments (Surgery -S-, R, C, CR, S + CR, S + R) came to the same conclusion (pooled HR 1.72 (95% CI 1.08–2.72), *p* = 0.02, for OS) (Figure 4B).

### 3.7. Impact of Cut-Off Value of NLR on Survival in All Patients

When the 14 studies were divided into groups according to a mean NLR cut-off value of 3.6, seven were included in the low NLR cut-off value group [13,14,17,19,20,21,26] and five in the high NLR cut-off value group [15,18,23,24,25]. The subgroup analysis showed that a high NLR was associated with lower OS in both groups, although the high cut-off subgroup had a slightly higher HR than the low cut-off subgroup (HR 2.09 (95% CI 1.46–2.98), *p* < 0.0001 and HR 1.47 (95% CI 1.08–2.00), *p* = 0.01, respectively) (Figure 4C,D).

### 3.8. Published Status Bias Analysis

Funnel plots for all studies of HRs for OS, DFS and RFS to examine this bias are illustrated in Appendix A. The plots show apparent asymmetry, indicative of bias, with fewer small studies reporting negative results than would be expected.

## 4. Discussion

This is the first meta-analysis specifically focusing on the prognostic significance of NLR in OPSCC. It includes a large number of patients (7647) from 14 studies and shows that OPSCC patients with a high NLR had a significant survival disadvantage, both in terms of OS and DFS, irrespective of the type of treatment. However, when the HPV status was considered, NLR was associated with survival only in HPV-positive patients. It is observed that the number of studies in which HPV status was considered is fairly limited (389 patients from three studies) and this may have influenced these results.

NLR has been regarded as of prognostic significance in multiple tumor types [45], with a high NLR value consistently being associated with a worse prognosis [46,47]. Previous meta-analyses including all HNSCC locations also have shown such a relationship [19,39,40,41,42]. The relationship was also shown in specific locations such as larynx [9,48] and oral cavity [49,50]. In accordance with these the present meta-analysis indicates that a high NLR is associated with a shortened survival (OS and DFS) in OPSCC patients.

A high NLR is associated with systemic inflammation. It has been shown that cancer cells recruit and activate neutrophils, called tumor-associated neutrophils (TAN); although their precise role within the tumor microenvironment remains controversial, they appear to facilitate tumor progression contributing to tumor growth, angiogenesis, immune tolerance and metastatic spread [51,52]. Studies of laryngeal squamous cell carcinomas have shown that TAN effect decreased CD4^+^/CD8^+^ T cell and inhibit production of TNF-α and IFN-γ, resulting in an immunosuppressive environment [53]. In addition, recruited and activated neutrophils contribute to the destruction of basement membranes, thereby facilitating stromal and lymphovascular invasion [54]. Thus, while an immune response to the tumor, manifested with predominance of lymphocytes within the tumor microenvironment, and possibly reflected in a low NLR, is beneficial to the patient, an inflammatory response, with predominance of neutrophils within the tumor microenvironment and an elevated NLR, could be harmful. However, no previous studies have addressed the neutrophil-to-lymphocyte ratio (for example, TAN-to-CD8^+^ ratio) and its prognostic significance in tumor tissues. Additionally, caution should be exerted and the role of tumor ulceration as a means of effecting an acute inflammatory response needs to be further assessed.

When subgroup analysis for HPV status was performed, the association of a high NLR with a poorer survival was only observed in the HPV-positive group of patients. Recent studies indicate differences in the host immune responses and populations of immune cells in the tumor microenvironment between HPV-negative and HPV-positive OPSCC patients [55,56]. Previous studies that stratified the results of the NLR by HPV status also indicated this. Brewczyński et al. [20] showed that the mean NLR was lower in patients with HPV-positive OPSCC (2.31) than in the HPV-negative ones (2.71), and that a higher NLR prior to treatment was a significant poor prognostic factor for both OS and DFS in the HPV-positive group but not in the patients with an HPV-negative tumor. Additionally, Radichi et al. [29], in their study including tumors from all head and neck sites, showed that HPV-positive patients had a lower mean NLR (2.73) than the HPV-negative patients (4.75), and that the increase in the risk of death with elevated NLR was greater in HPV-positive patients compared to HPV-negative ones. Huang et al. [33] studied the prognostic significance of circulating neutrophils, monocytes, and lymphocytes in OPSCC stratified by HPV status. They found that although HPV-positive patients had lower neutrophil and monocyte counts compared to HPV-negative patients, both elevated circulating neutrophil and monocyte counts were associated with decreased OS only in HPV-positive patients.

Taken together, the present meta-analysis and the aforementioned studies suggest that the effect of elevated NLR on prognosis is greater in HPV-positive than in HPV-negative OPSCC. Therefore, stratification according to HPV status is recommended for future studies.

There is no established cut-off value for the NLR, pertinent differences in the included studies ranging between two and five (mean 3.6, median 3.4). The cut-off NLR was established in most studies using a ROC curve (6/14 studies) or median values of their respective cohort (3/14 studies). We divided the studies into two categories of cut-off values based on the mean value (NLR cut-off values < 3.6 or ≥3.6) and performed a meta-analysis for each subgroup. The results suggest that elevated NLR relates to a higher mortality risk in the subgroup of studies that used a higher cut-off value; accordingly, the use of higher cut-off values may be of more prognostic significance. 

The present meta-analysis indicates an association between a high NLR prior to treatment and shorter OS and DFS in OPSCC, but there are still several limitations. Firstly, Appendix A indicate the presence of a published bias in the analysis, and the possibility of unpublished data cannot be excluded. Secondly, all included studies were retrospective studies with a nonrandomized design, which might enhance the influence of confounders. Thirdly, the HPV status, was only considered in a minority of the studies. The *I*^2^ for all studies pooled was 91%, indicating high heterogeneity; but this heterogeneity was substantially reduced when the subgroup analysis by HPV status was performed (0% in HPV-positive group and 57% in HPV-negative group for OS); this seems reinforcing the importance of HPV status in the prognostic significance of NLR. Finally, it should be appreciated that while NLR is a useful biomarker and systematic assessments, as here, are of interest, the problem always lies with the relevance to an individual patient. While globally NLR allows some inference about the ‘inflamed state’ in an individual, it may simply reflect a mere infection.

## 5. Conclusions

The present meta-analysis indicates the prognostic significance of NLR value in the largest reported cohort of OPSCC patients. High NLR was related to a worse clinical outcome in OPSCCs, specifically in the HPV-positive subgroup. Different treatment modalities and NLR cut-off values did not affect the relation between high NLR and OS. Accordingly, the assessment of NLR value in peripheral blood prior to treatment of OPSCC seems worthy of further investigation to assess its role as a standardized and validated biomarker in prospective clinical trials.

## Figures and Tables

**Figure 1 cancers-15-00802-f001:**
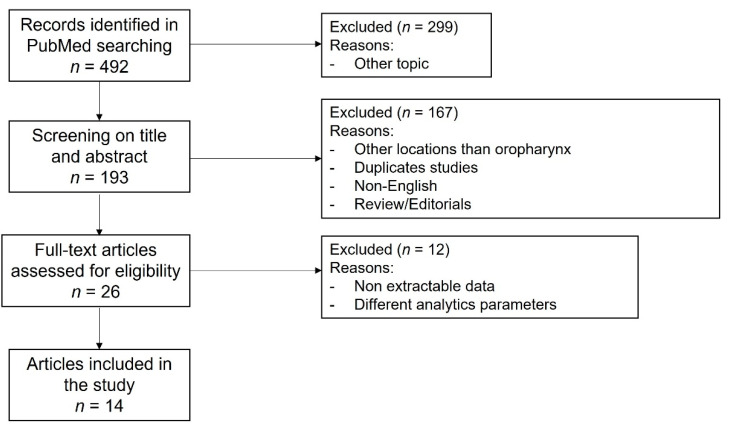
Flow chart showing the process of the study selections for the systematic review.

**Figure 2 cancers-15-00802-f002:**
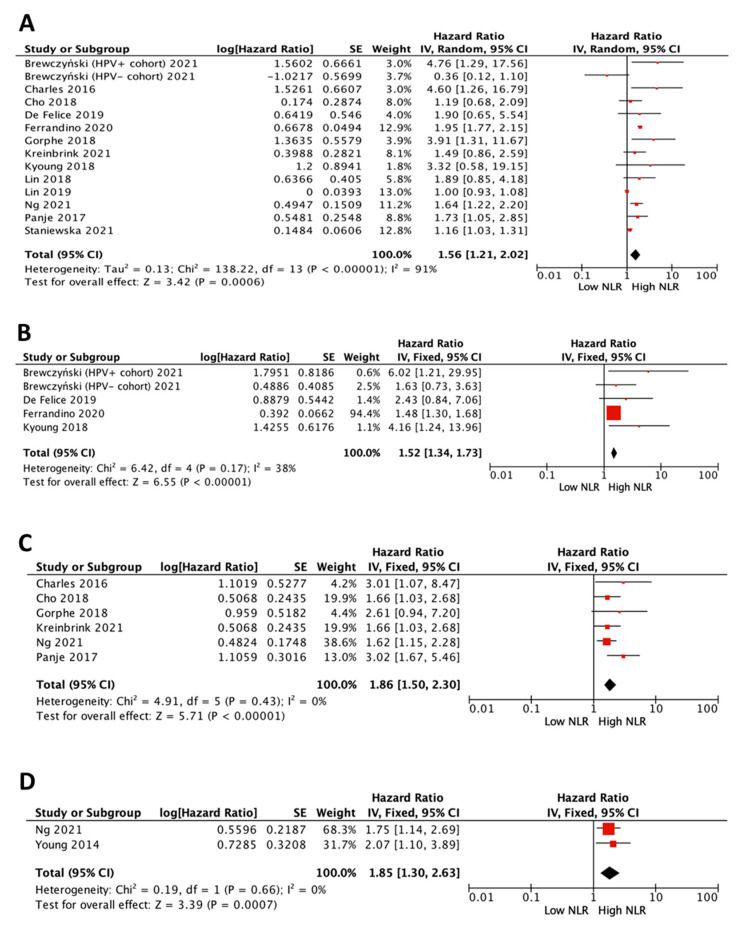
Forest plots of impact of NLR in “all patients” on overall survival (**A**), disease-free survival (**B**), recurrence-free survival (**C**) and locoregional control (**D**). [13,14,15,16,17,18,19,20,21,22,23,24,25,26].

**Figure 3 cancers-15-00802-f003:**
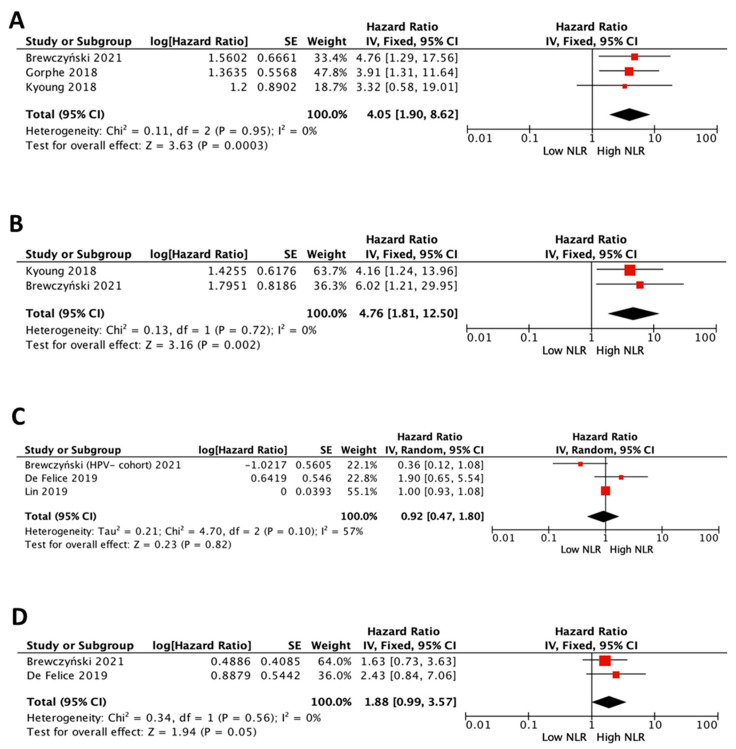
Forest plots of impact of NLR in “HPV+ patients” on overall survival (**A**) and disease-free survival (**B**), and of the impact of NLR in “HPV- patients” on overall survival (**C**) and disease-free survival (**D**). [14,15,18,20,25].

**Figure 4 cancers-15-00802-f004:**
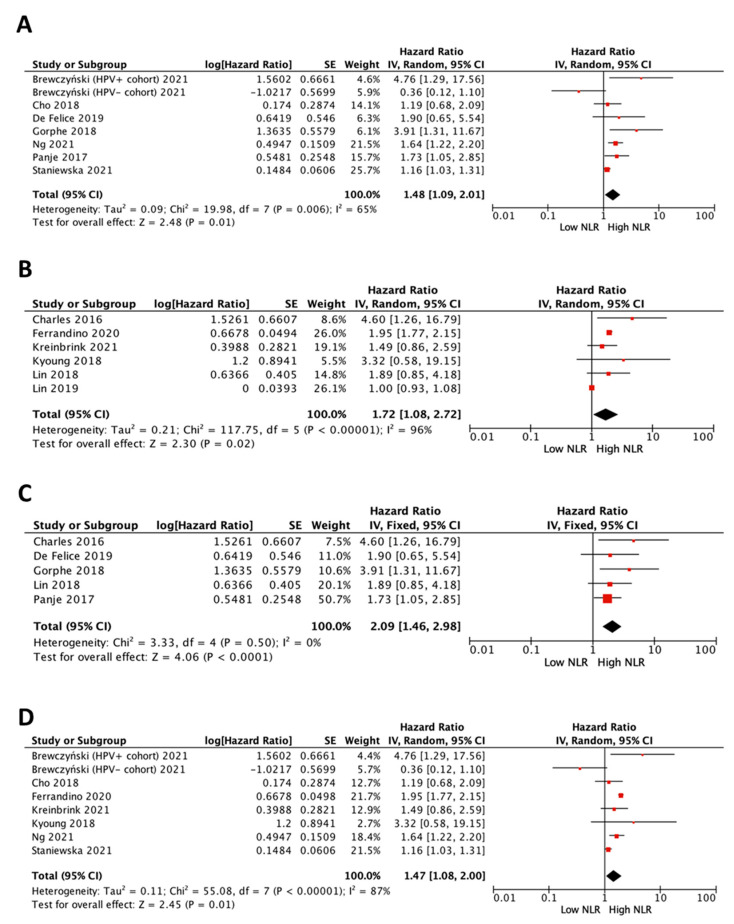
Forest plots of impact of NLR in “all patients” on overall survival with radiotherapy or chemoradiotherapy treatment (**A**) and mixed primary treatment (**B**), and in patients in the group of NLR cut-off ≥ 3.6 (**C**) and NLR cut-off < 3.6 (**D**). [13,14,15,16,17,18,19,20,21,23,24,25,26].

**Table 1 cancers-15-00802-t001:** Main features of the selected studies.

References	Year	Country	Sample Size	Clinical Stage	Median Age	Median Follow-Up	HPV Status
Ferrandino et al. [13]	2020	United States	5169	I, II, III, IV	64	60	All patients
So et al. [14]	2018	Republic of Korea	104	I, II, III, IV	57	60	Positive
De Felice et al. [15]	2019	Italy	57	I, II, III, IV	NA	60	Negative
Lin et al. [16]	2019	United States	108	I, II, III, IV	56	37	Negative
Kreinbrink et al. [17]	2021	United States	201	I, II, III, IV	58	40	All patients
Lin et al. [18]	2018	United States	99	I, II, III, IV	54	69.6	All patients
Cho et al. [19]	2018	Republic of Korea	56	I, II, III, IV	60	39	All patients
Brewczyński et al. [20]	2021	Poland	127	I, II, III, IV	61	74.58	Positive and negative
Staniewska et al. [21]	2021	Poland	208	I, II, III, IV	59	NA	All patients
Young et al. [22]	2014	United Kingdom	249	I, II, III, IV	55	46	All patients
Charles et al. [23]	2016	Australia	67	I, II, III, IV	63	29	All patients
Panje et al. [24]	2017	Switzerland	187	I, II, III, IV	62	61.2	All patients
Gorphe et al. [25]	2018	France	167	I, II, III	59	32	Positive
Ng et al. [26]	2021	United States	848	II, III, IV	57	59	All patients
**References**	**Treatment Method**	**Covariant**	**Cut-Off Value of NLR**	**Method of Obtaining the Cut-Off Value**	**Survival Analysis**	**Analysis Method**	**NOS**
Ferrandino et al. [13]	S, R, C, CR, S + CR, S + R	Top tertile NLR	3.4 (OS) 3.7 (DFS)	Tertile	OS, DFS	M	9
So et al. [14]	S, R, CR, S + CR, S + R	High NLR	2.42	ROC	OS, DFS	M	8
De Felice et al. [15]	R, CR	High NLR	4.7	ROC	OS, DFS	M	9
Lin et al. [16]	S + R, R, CR	High NLR	NA	Median	OS	U	8
Kreinbrink et al. [17]	S + R, R, CR	High NLR	3.0	Median	OS, RFS	U	8
Lin et al. [18]	S + R, S + CR	High NLR	3.8	ROC	OS	U	9
Cho et al. [19]	R, CR	High NLR	2.7	ROC	OS, RFS	M/U	9
Brewczyński et al. [20]	R, CR	High NLR	2.13 (OS) 2.29 (DFS)	ROC	OS, DFS	M/U	9
Staniewska et al. [21]	R, CR	High NLR	2.009	ROC	OS	M	7
Young et al. [22]	CR	High NLR	5	NA	LRC	M	7
Charles et al. [23]	S + R, S + CR, R, CR	High NLR	5	Review of literature	OS, RFS	M	8
Panje et al. [24]	R, CR	Top quartile NLR	4.68	Quartile	OS, RFS	M	9
Gorphe et al. [25]	CR	High NLR	5	Review of literature	OS, RFS	M	7
Ng et al. [26]	R, CR	High NLR	3.0	Median	OS, RFS, LRC	M	9

NOS: Newcastle-Ottawa scale; S: Surgery; R: Radiotherapy; C: Chemotherapy; CR: Chemoradiation; NLR: neutrophil-to-lymphocyte ratio; OS: Overall survival; DFS: Disease-free survival; RFS: Recurrence-free survival; LRC: Loco-regional control; M: Multivariate; U: Univariate; ROC: Receiver-operating characteristics curve.

## Data Availability

The datasets generated during and/or analyzed during the current study are available from the corresponding author on reasonable request.

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
