# Peer review of "Neutrophil to Lymphocyte Ratio in Oropharyngeal Squamous Cell Carcinoma: A Systematic Review and Meta-Analysisâ€"

_cancers, 2023, doi:10.3390/cancers15030802_

Round 1
Reviewer 1 Report
cancers-2087654
Title: “Neutrophil to lymphocyte ratio in oropharyngeal squamous cell carcinoma: a systematic review and meta‑analysis"
Authors: Rodrigo et al.
Here, the authors presented a meta-analysis analyzing the clinical relevance of neutrophil-to-lymphocyte ratio (NLR) in oropharyngeal squamous cell carcinoma (OPSCC). Overall, 14 studies including n= 7,647 patients were analyzed concerning their NLR, and survival rates. Additionally, patients were sub-grouped according to their HPV status. In summary, the authors concluded that an increased NLR correlated with worse prognosis, especially in HPV-positive OPSCC.
Clearly, the authors’ work is appreciated aiming at the revision of a clinical relevant, “easy accessible” biomarker for the prognosis of OPSCC. However, in my opinion the scope of the study is limited focusing only on a limited subgroup of HNSCC. There are several previous reports about the topic for HNSCC (eg [1-5]), so novelty must be evaluated critically.
Additionally, I feel that the authors can improve the manuscript presentation. In summary, I am unfortunately unable not recommend acceptance of the report for publication in Cancers in its current form. Maybe, the authors should re-submit to a more specialized journal of the MDPI group.
Some general minor points should be addressed before publication.
- abstract should be revised also providing general conclusions, not only giving numbers!
- introduction is very short and needs further details, eg about the characteristics of HPV-driven OPSCC since this is one major focus of the study.
- presentation of the results can be improved, especially of the figures. Of course, meta-analyses always contain a lot of “numbers”, but summarizing the results in one “TOC figure” would greatly increase the interest to the reader!
References
1. Mariani, P.; Russo, D.; Maisto, M.; Troiano, G.; Caponio, V.C.A.; Annunziata, M.; Laino, L. Pre-treatment neutrophil-to-lymphocyte ratio is an independent prognostic factor in head and neck squamous cell carcinoma: Meta-analysis and trial sequential analysis. J Oral Pathol Med 2022, 51, 39-51, doi:10.1111/jop.13264.
2. Pan, C.; Wu, Q.V.; Voutsinas, J.; Houlton, J.J.; Barber, B.; Futran, N.; Laramore, G.E.; Liao, J.J.; Parvathaneni, U.; Martins, R.G., et al. Neutrophil to lymphocyte ratio and peripheral blood biomarkers correlate with survival outcomes but not response among head and neck and salivary cancer treated with pembrolizumab and vorinostat. Head & neck 2023, 45, 391-397, doi:10.1002/hed.27252.
3. Takenaka, Y.; Oya, R.; Takemoto, N.; Inohara, H. Neutrophil-to-lymphocyte ratio as a prognostic marker for head and neck squamous cell carcinoma treated with immune checkpoint inhibitors: Meta-analysis. Head & neck 2022, 44, 1237-1245, doi:10.1002/hed.26997.
4. Yanni, A.; Buset, T.; Bouland, C.; Loeb, I.; Lechien, J.R.; Rodriguez, A.; Journe, F.; Saussez, S.; Dequanter, D. Neutrophil-to-lymphocyte ratio as a prognostic marker for head and neck cancer with lung metastasis: a retrospective study. Eur Arch Otorhinolaryngol 2022, 279, 4103-4111, doi:10.1007/s00405-022-07274-1.
5. Zhao, Y.; Qin, J.; Qiu, Z.; Guo, J.; Chang, W. Prognostic role of neutrophil-to-lymphocyte ratio to laryngeal squamous cell carcinoma: a meta-analysis. Braz J Otorhinolaryngol 2022, 88, 717-724, doi:10.1016/j.bjorl.2020.09.015.
Author Response
Reviewer #1
- Comment #1
“Clearly, the authors’ work is appreciated aiming at the revision of a clinical relevant, “easy accessible” biomarker for the prognosis of OPSCC. However, in my opinion the scope of the study is limited focusing only on a limited subgroup of HNSCC. There are several previous reports about the topic for HNSCC (eg [1-5]), so novelty must be evaluated critically”.
Response to the comment:
The reviewer notes that the scope of the paper is limited by focusing only on a subgroup of head and neck tumors. However, this was precisely our objective and what we consider one of the strengths of the study: there are already multiple studies (as shown in the introduction and in the literature provided by the reviewer) that point out the prognostic value of the NLR, but as the different head and neck sublocations have particularities in terms of biological and prognostic behavior, it is not known if this prognostic value is equally applicable to all locations or if it can be influenced by the type of treatment. For this reason, in this work we have focused on a specific sublocation (oropharynx), which also has the particularity that a part of the tumors originating in it are produced by HPV. We have not only focused on this specific localization, but we have also performed an analysis according to the HPV status and the type of treatment received by the patients (this type of meta-analysis has not been performed previously). Our results show that the NLR has prognostic value only in the subgroup of HPV-positive patients, which proves that our approach is correct. Had we performed an analysis without focusing on oropharyngeal carcinomas and taking into account HPV status, we would not have been able to detect these differences in the prognostic value of the NLR.
- Comment #2
“Additionally, I feel that the authors can improve the manuscript presentation. In summary, I am unfortunately unable not recommend acceptance of the report for publication in Cancers in its current form. Maybe, the authors should re-submit to a more specialized journal of the MDPI group.”
Response to the comment:
The reviewer suggests referring the article to a more specialized journal within the MDPI group: precisely, the article was submitted to a special issue dedicated to "The biomarkers and detection of head and neck cancer", fitting well within the scope of this special issue.
- Minor issues
- “abstract should be revised also providing general conclusions, not only giving numbers!”
The abstract was revised according to these recommendations.
- “Introduction is very short and needs further details, eg about the characteristics of HPV-driven OPSCC since this is one major focus of the study.”
The introduction was augmented by including some more information about HPV-driven OPSCC and its differences with HPV-negative OPSCC.
- “Presentation of the results can be improved, especially of the figures. Of course, meta-analyses always contain a lot of “numbers”, but summarizing the results in one “TOC figure” would greatly increase the interest to the reader!”
Regarding the presentation of the results, I understand that the figures showing the results of the different meta-analyses performed can be cumbersome, but it is the correct and standard way to show the analyzed data so that they can be critically evaluated by the readers. However, following the reviewer's recommendation, we have included a summary figure in the form of a graphical abstract.
Reviewer 2 Report
The submitted systematic literature review concerns an important topic and displays the state-of-art literature background for the relation of the neutrophil-to-lymphocyte ratio (NLR) in peripheral blood, which is regarded as the balance between systemic inflammation and immunity, with oropharynx cancer taking the HPV background also extra into consideration. The main outcome was that high NLR was associated with worse survival.
Comments
The Introduction is appropriate and contains the required backgound information.
Methods
The used PRISMA analysis method and the subsequent process are correct and well documented.
Results
The results are presented in appropriate form.
Discussion
One optional question
In the Discussion it is mentioned that “Tumor-associated neutrophils (TAN) seem to contribute to tumor growth, angiogenesis, immune tolerance and metastatic spread [52].” In the meta-analysis the systematic neutrophil-to-lymphocyte ratio was investigated. Are studies available for the local in tumor tissue investigated tumor-associated neutrophil/lymphocyte ratio, and do these show differences? This question also considers that a non-systematic relative low extent local neutrophil load in the tumor tissue does have a role or not in the outcome?
Author Response
Comments:
“The Introduction is appropriate and contains the required backgound information.
Methods
The used PRISMA analysis method and the subsequent process are correct and well documented.
Results
The results are presented in appropriate form.”
Response to the comments:
Thank you for your favorable comments
Minor issues:
“One optional question
In the Discussion it is mentioned that “Tumor-associated neutrophils (TAN) seem to contribute to tumor growth, angiogenesis, immune tolerance and metastatic spread [52].” In the meta-analysis the systematic neutrophil-to-lymphocyte ratio was investigated. Are studies available for the local in tumor tissue investigated tumor-associated neutrophil/lymphocyte ratio, and do these show differences? This question also considers that a non-systematic relative low extent local neutrophil load in the tumor tissue does have a role or not in the outcome?”
As we mention in the discussion, it has been shown that TAN decreases CD4+/CD8+ T cells in tumor tissues, but the ratio between neutrophils (TAN) and lymphocytes (i.e. CD8+ cells) in tumor tissues and its prognostic significance has not been studied. This is now mentioned in the discussion.